# Phyto-Synthesis, Characterization, and In Vitro Antibacterial Activity of Silver Nanoparticles Using Various Plant Extracts

**DOI:** 10.3390/bioengineering9120779

**Published:** 2022-12-07

**Authors:** Bilal Ahmad, Li Chang, Usama Qamar Satti, Sami ur Rehman, Huma Arshad, Ghazala Mustafa, Uzma Shaukat, Fenghua Wang, Chunyi Tong

**Affiliations:** 1College of Biology, Hunan University, Changsha 410082, China; 2College of Chemistry and Chemical Engineering, Hunan University, Changsha 410082, China; 3Institute of Bast Fiber Crop, Chinese Academy of Agriculture Sciences, Changsha 410082, China; 4Key State Laboratory of Mechanical Behaviour of Materials, Department of Materials Science and Engineering, Xi’an Jiaotong University, Xi’an 710049, China; 5Department of Materials Science and Engineering, Institute of Space Technology, Islamabad 44000, Pakistan; 6Department of Plant Sciences, Quaid-i-Azam University, Islamabad 44000, Pakistan; 7National Institute of Health (NIH) Park Road, Chak Shahzad, Islamabad 45550, Pakistan; 8Institute of Physical Education, Xinjiang Normal University, 102 Xinyi Rd., Urumqi 830054, China

**Keywords:** Ag-NPs, green synthesis, *Aloe vera*, anti-bacterial, phytotoxicity

## Abstract

*Aloe vera*, *Mentha arvensis* (mint), *Coriandrum sativum* (coriander), and *Cymbopogon citratus* (lemongrass) leaf extracts were used to synthesize stable silver nanoparticles (Ag-NPs) by green chemistry. UV–vis spectrophotometry, X-ray diffraction (XRD), thermogravimetric analysis (TGA), scanning electron microscopy (SEM), and energy dispersive X-ray (EDX) spectroscopy techniques were used to characterize these biosynthesized nanoparticles. The data indicated that the silver nanoparticles were successfully synthesized, and the narrower particle size distribution was at 10–22 nm by maintaining a specific pH. As a short-term post-sowing treatment, Ag-NP solutions of different sizes (10 and 50 ppm) were introduced to mung bean seedlings, and the overall increase in plant growth was found to be more pronounced at 50 ppm concentration. The antibacterial activity of Ag-NPs was also investigated by disc diffusion test, minimum inhibitory concentration (MIC), and minimum bactericidal concentration (MBC) test. The zones of inhibition (ZOI) were shown by *Escherichia coli* (*E. coli*) (1.9, 2.1, 1.7, and 2 mm), followed by *Staphylococcus aureus* (*S. aureus*) (1.8, 1.7, 1.6, and 1.9 mm), against coriander, mint, *Aloe vera*, and lemongrass, respectively. MIC and MBC values of *E. coli,* and *S. aureus* ranged from 7 to 8 µg/mL. Overall, this study demonstrates that Ag-NPs exhibit a strong antimicrobial activity and thus might be developed as a new type of antimicrobial agent for the treatment of bacterial infection.

## 1. Introduction

Since its inception more than 20 years ago, nanotechnology has had a significant impact on the environment, human health, and society. It is a dynamic, revolutionary, and rapidly growing multidisciplinary area that works at a nanoscopic scale (10^−9^ m). When compared to their equivalent materials, nanomaterials (whether natural, engineered, or incidental) such as nanomaterials typically exhibit dimensions of below 100 nm along with unique physical, chemical, and biological properties (such as a high surface-to-volume ratio, surface functionalization, controlled targeting, and release). They are highly appealing tools due to their potential applications in a variety of industries, such as cosmetics, pharmacy, biotechnology, chemistry, and agriculture [1].

Nanoparticles have attracted much attention due to their different properties, including size, shape, optical, magnetic, and electrical properties [2]. They can be used to fuse fibers, biosensors, and electromechanics, and their anticancer properties have pharmaceutical potential [3,4,5,6]. As there is a growing desire to manufacture nanoparticles using physiologically friendly processes, nanomaterials synthesis has recently become one of the most fascinating technological fields of study [7]. Although there are various methods for synthesizing nanomaterials, most of them are physical or chemical. One example is the use of green chemistry: biosynthetic approaches to solve problems with toxicity, cost, and manufacturing time [8]. Other methods, although superior to biological methods, are more expensive and produce toxic nanomaterials, limiting their application in the medical industry [9]. As a result, developing safe, clean, cost-effective, and biocompatible nanomaterials with environmentally friendly methods is critical. These biological methods are safe and environmentally friendly, and they do not necessitate any preparatory conditions or processes [10].

Because of their distinctive features (physical, chemical, and biological), and because of their diverse demand in sectors such as electronics, optics, and medicine, silver nanoparticles (Ag-NPs) have gained substantial attention in the field of metallic nanoparticle research [11]. These nanomaterials are used as antiseptics in manufacturing industries, health care products, medical device covers, optical devices, drug delivery, and as anti-cancer agents [12]. Moreover, these particles are also used in water treatment, agriculture, food packaging, and in the textile industry, wound-dressing, drug delivery, biosensors, orthopedics, and medical diagnostics [13]. Various biological activities of Ag-NPs, such as their anti-bacterial, antiviral, antifungal, antioxidant, anti-cancer, anti-mycobacterial, anti-platelet, and anti-diabetic activities, have been well-reported in several earlier studies [14,15]. 

The manufacture of these metallic nanomaterials has grown in relevance in recent years due to their exceptional capabilities, such as their catalytic, optical, and electrical qualities [16,17]. Because of their antibacterial action and the ease with which their salts may be decreased, biosynthesis of these metallic nanomaterials using green synthesis is becoming more popular [18,19]. This is a simple, one-step approach that is ideal for large-scale production. It is also a less expensive and more environmentally friendly technology, which is why it is employed in medical research [20]. 

*Mentha*, a sterile hybrid of the species *M. aquatica* L. and *M. spicata*, is likely the most significant commercial aromatic herb in the world today. Mint leaves have a distinctive, sweet-smelling, potent flavor that is warm and spicy with a cooling aftertaste [21]. Mint leaves have antimicrobial and antioxidant qualities that can help prevent cancer, diabetes, and vascular issues [22]. The monoterpenes limonene, isomenthone, menthol, menthofuran, D-neoisomenthol, 1,8-cineole (eucalyptol), D-carvone, linalool, linalyl acetate, piperitenone oxide, and pulegone are present in higher concentrations in *Mentha*. Researchers have found that mint leaves and their parts have health benefits, such as protecting the heart from ischemia-reperfusion, helping the kidneys get rid of urate, and helping with high blood pressure, diabetes, and high cholesterol [23]. 

*Aloe vera*, a succulent, is also considered to be a wonder plant [24]. This plant has long been used to make herbal medications. *Aloe vera* plant extracts are utilized in a variety of industrial uses in addition to medicine. For example, they are employed in calming applications. *Aloe vera* extracts are commonly used in cosmetics because they offer therapeutic properties. *Aloe vera*’s safety has been scientifically proven, and it therefore has a wide range of applications in the biomedical field [25]. Polysaccharides, flavonoids, carbohydrates, coumarins, tannins, chromones, alkaloids, anthraquinones, organic compounds, pyrones, phytosterols, anthrones, sterols, vitamins, proteins, and mineral components are among the phytochemical components of *Aloe vera* [26]. 

Lemongrass (Cymbopogon genus) is a perennial grass in the Poaceae family that grows tall and in clumps. The Indian subcontinent’s tropical and subtropical regions, South and North America, Africa, Australia, and Europe are the primary regions where this genus is found. It has more than 55 species, but the two most prevalent are *Cymbopogon flexuosus* and *Cymbopogon citratus*. Due to its antiseptic, antibacterial, antimicrobial, antifungal, and anti-inflammatory characteristics, this plant is widely used in pharmaceutical processes, as well as in agriculture and therapeutic uses. Lemongrass can also be used as a starting ingredient in the manufacture of biogas and silica [27]. 

The herb *Coriandrum sativum*, often known as coriander, is an herb whose entire plant is edible [28]. It can be found throughout Europe, North Africa, and Southwest Asia, among other places. The plant is used in a variety of cuisines around the world [29]. It is frequently utilized in various treatments for the treatment of upset stomach, joint pain, worms, nausea, and diarrhea because it is a medicinal plant. Coriander plants grow to a height of 50 cm and have a gentle growth habit. Its fruit has a diameter of 3–5 mm on average. Coriander also has antibacterial activity against gram-positive and gram-negative bacteria such as *E. coli*, *P. aeruginosa*, and *E. faecalis* [28]. 

In the present study, four different plant extracts were used for the synthesis of Ag-NPs. We used a green approach for the synthesis of nanomaterials to develop a simple, ecofriendly, and rapid method in which plant extracts were used as capping and stabilizing agents. The nanomaterials were characterized using various modern techniques. Although several studies have been carried out that involved the green synthesis and characterization of nanomaterials, there are nonetheless few reports on the comparison of various plant extracts in the synthesizing process and their subsequent effect on the size and morphology of the synthesized nanomaterials. This study not only explores the effects caused by the nature of the plant extracts on the synthesis but also explores the bioactivity of these nanomaterials. Afterwards, the phytotoxic potential of the synthesized nanomaterials was also assessed on mung bean seedlings.

## 2. Materials and Methods 

### 2.1. Materials

Silver nitrate (AgNO_3_) procured from Sigma-Aldrich (St. Louis, MO, USA) *was* used as a precursor for the synthesis of silver nanoparticles. Ethanol was also procured from Sigma-Aldrich, which was used in the washing of silver nanoparticles. All the plant leaves, i.e., mint, coriander, *Aloe vera*, and lemongrass, were taken from the Biological Garden of Quaid-i-Azam University Islamabad, Pakistan and College of Biology, Hunan University, Changsha, China.

### 2.2. Preparation of Plant Extracts

Fresh mint leaves (25 g) were separated, washed thoroughly, dried, and then crushed in 100 mL of double-distilled, sterilized water and allowed to boil for 30 min at 300 °C. The mint extract thus formed was allowed to cool, and then it was filtered three times via Whatmann filter paper No. 1, which has a pore size of 11 µm. The extract was then stored at 4 °C for Ag-NP synthesis [30]. For the preparation of *Aloe vera* plant extract, 100 g leaves were selected, washed, and then dried in an oven at 35 °C for 3 h. After that, they were boiled in 500 mL of double-distilled purified water for 90 min at 300 °C. The *Aloe vera* extract was allowed to cool at room temperature, and then it was filtered 3 times via Whatmann filter paper No. 1 and stored at 4 °C for nanoparticle synthesis [31]. 

Fresh lemongrass leaves (50 g) were selected, finely cut, washed thoroughly, and boiled in 200 mL of double-distilled sterilized water for 30 min at 300 °C. The extract obtained was allowed to cool at room temperature, after which it was filtered three times with Whatmann filter paper No. 1. The extract was kept at 4 °C for more nanoparticle synthesis [32]. Fresh coriander leaves (23 g) were selected and boiled in 100 mL of double-distilled sterilized water at 300 °C for 30 min after thorough washing. The extract was then filtered 3 times with Whatmann filter paper No. 1 and stored at 4 °C for further experiments [33].

### 2.3. Synthesis of Silver Nanoparticles

The use of several toxic chemical reducing agents and dispersants was avoided during our environmentally friendly and green synthesis of silver nanoparticles. In addition to being a novel means to fully utilize plant extracts, our method also represents a novel approach to the synthesis of silver nanoparticles by a green method. The schematic illustration of Ag-NPs, which were synthesized through plant extracts, is shown in Figure 1.

### 2.4. Characterizations of Silver Nanoparticles

The synthesized nanoparticles were characterized using field emission scanning electron microscopy (FESEM) and MIRA TESKAN at 10 kV for the morphology investigation and size estimation. EDS analysis was performed for the elemental analysis to confirm the formation of Ag-NPs produced via different plant extracts. Using distilled water, MALVERN Master Sizer Hydro 3000 equipment was used for particle size analysis. Later, the development and morphology of Ag-NPs were confirmed. Their size distributions were also assessed. XRD was performed and used to find the crystallographic structure of the Ag-NPs. The carbon X-rays were introduced with a wavelength of 1.5–1.6 Å. Thermogravimetric analysis was performed on Mettler Toledo (TGA/SDTA851) to evaluate the purity of Ag-NPs. The TGA analysis was performed in the air from 30 to 800 °C at a rate of 10 °C per min.

### 2.5. Comparative Study of Mung Bean (Vigna radiata L.) Responses towards Ag-NPs and Their Salt

Experimentation was done to find the effect of Ag-NPs and AgSO_4_ on mung beans (Azri mung 2018) with two concentrations (10 mg/L = T1 and 50 mg/L = T2) of nanoparticles and salt, along with control (0 mg/L = T0). Stress was applied on the sixth day after germination at the two-leaf stage. Morphological parameters, i.e., fresh plant weight (root weight, shoot weight) and whole plant length were observed for each treatment (10 mg/L = T1, 50 mg/L = T2 and 0 mg/L = T0). Healthy seeds of *Vigna radiata* L. (Azri Mung 2018) were collected from the Crop Science Institute, NARC (National Agriculture Research Center), Islamabad, Pakistan.

### 2.6. Seed Sterilization and Germination

Seeds were surface-sterilized for 3 min with a (3% *v*/*v*) sodium hypocrite solution, and sterilized seeds were washed four times (two-minute wash each time) with distilled water. Seeds were sown in 800 mL of pure/blank sand and mixed well with 250 mL of water. Twenty seeds were sown equidistantly per pot (8 × 6 inches). The properly labeled pots were then shifted into a growth chamber with 47% humidity and a 16 ± 8 h light–dark period and a constant temperature of 25 °C.

### 2.7. Treatments

The selected concentrations of silver salt solutions and silver nanoparticle suspensions were applied at two leaf stages of the plant (6th day after sowing). Each pot received 5 mL of each treatment (0 mg/L = T0, 10 mg/L = T1, and 50 mg/L = T2). Data were collected for three consecutive days: Day 1, after applying stress (D1); Day 2, after applying stress (D2); and Day 3, after applying stress (D3). The effects of each treatment were compared, and data were collected for each replicate. All the experiments were performed in triplicate. A toxicity study was carried out on mung bean seedlings. Nanoparticle stress was applied on the 6th day of germination and measurements were taken for the next three consecutive days, i.e., D1 is Day 1 after stress application, which was the 7th day post-germination. 

### 2.8. Plant Length

The root and shoot lengths of mung bean plants (Azri Mung 2018) were measured using a measuring scale in cm on the 7th, 8th, and 9th days after sowing for the control group and groups treated with different concentrations of Ag-NPs and AgSO_4_ (0 mg/L, 10 mg/L, and 50 mg/L).

### 2.9. Root and Shoot Weight

The fresh weight of mung bean plant roots and shoots was measured using an electrical weighing balance (mg). Data were collected on the 7th, 8th, and 9th days after sowing under T0, T1, and T2 for Ag-NPs and AgSO_4_ with constant external conditions.

### 2.10. In Vitro Susceptibility, Minimum Inhibitory Concentration (MIC), and Minimum Bactericidal Concentration (MBC) Evaluation of Ag-NPs

Using the disc diffusion method, the antibacterial activity of biosynthesized Ag-NPs and AgNO3 solution against *E. coli* and *Staphylococcus aureus* were investigated. The antibacterial activity of Ag-NPs against the selected gram-negative and gram-positive foodborne pathogens was carried out using the Kirby–Bauer disk diffusion susceptibility test method [34]. The bacterial strains were spread on Mueller–Hinton agar (MHA) (Merck, Germany) using sterile cotton swabs. A sterile blank antimicrobial susceptibility disk was used in the test. Samples were then poured into the wells. One well was allocated for control, into which we poured phosphate buffer saline (PBS) as a control solution. Then, plant extract was poured into the well of each Petri dish. The disks were then placed on the agar plate and incubated at 37 °C for 24 h. Zones of inhibition were observed after 24 h of incubation.

The MIC and MBC of green-synthesized Ag-NPs were determined using the method described in the CLSI guidelines [35]. The MIC test was conducted in 96-well round-bottom microtiter plates using standard broth microdilution methods, while the MBC test was performed on the MHA plates. The MIC value was defined as the lowest concentration of antibacterial agents that inhibited the growth of bacteria. The MBC was defined as the lowest concentration of the antibacterial agents that completely killed the bacteria. The MBC test was performed by plating the suspension from each well of the microtiter plates into MHA plates. The plates were incubated at 37 °C for 24 h. The MBC value was determined by the lowest concentration on the MHA plates that had no visible growths.

## 3. Results

### 3.1. UV–Visible Spectroscopy

The generated solutions were measured for absorbance at different wavelengths to check for the synthesis of Ag-NPs. The kinetics of reduction of aqueous silver ions during reaction with the leaf extracts broth could be easily followed via UV–vis–NIR spectroscopy. It is well-known that Ag-NPs exhibit reddish brown colors, which arise due to the excitation of surface plasmon resonance (SPR) in the Ag-NPs. Figure 1 shows the UV–vis–NIR spectra recorded from the aqueous silver leaf extracts of mint, *Aloe vera*, coriander, and lemongrass broth reaction medium as a function of time of reaction. It was observed that as the reaction proceeded and the pH of the solution increased to 9, the silver SPR band at ca. 390–470 nm for the leaf extracts broth steadily decreased in intensity. This band is indicative of the presence of nanoparticles in solution. In addition to the peaks in the abovementioned range, a progressive decrease in the absorption at smaller wavelengths into the near-infrared (NIR) region of the electromagnetic spectrum was observed.

The samples of prepared nanoparticles were subjected to the JASCO V-730 UV–Visible Spectrometer at the National Center for Physics in Islamabad for this purpose. The optimal UV–visible spectrum for silver nanoparticles is between 390 nm and 470 nm. All samples of green synthesized Ag-NPs showed their absorbance peaks within the range of the ideal spectrum of silver. This also indicated that the particles thus formed were silver nanoparticles. The Ag synthesized in this research work was in the form of suspended nanoparticles in DI ethanol. The SEM sample preparation was conducted without sonication to minimize the risk of impurities. The agglomeration is due to the adhesion of particles to each other by weak forces once the samples are prepared on carbon tape. The stages of agglomeration are particle movement, particle collision, particle attachment, and either disintegration or cementation of the attached particles.

### 3.2. Field Emission Scanning Electron Microscopy (FESEM)

Ag-NPs were synthesized biologically from mint, coriander, *Aloe vera*, and lemongrass leaf extracts. These Ag-NPs were affirmed using FESEM coupled with EDX, as shown in Figure 2. FESEM reveals that most biologically synthesized Ag-NPs are round. The synthesis of Ag-NPs was achieved by changing the concentration of plant extract in the solution of distilled water and plant extract as reported by similar routes [10,11,12,13]. Ag-NPs were added dropwise to the plant extract with continuous stirring until the solution reached pH 9. By this procedure, the relative size of the synthesized nanoparticles was reduced as compared to the particles synthesized at low pH, independent of the morphology produced. Figure 2 shows FESEM images of Ag-NPs synthesized from different plant extracts, in which the spherical morphology of NPs and the reduction in the relative size of NPs is visible as compared to the particles synthesized at low pH. The SEM analysis unambiguously presented the occurrence of Ag-NP synthesis. NPs ranging from 15 to 50 nm were found in the apparent shape of Ag-NPs. The smaller particles were almost spherical in shape, and some of them were aggregated (Figure 2). The SEM micrographs showed aggregates of synthesized Ag-NPs, and the particles were in the range of 15–25 nm and are in direct contact even within the aggregates, indicating the stability of NPs by controlling the pH.

### 3.3. Energy-Dispersive X-ray Spectroscopy (EDX)

Elemental analysis of Ag-NPs using EDX confirmed the existence of silver in larger concentrations. EDX also validated the nanoparticles under FESEM of silver. The EDX analysis of *Aloe vera*, lemongrass, mint and coriander are shown in Figure 3.

### 3.4. Particle Size Analysis (PSA)

PSA was performed on four samples of Ag-NPs by forming a suspension in distilled water. Ag-NPs produced using *Aloe vera* showed a narrower particle size distribution as compared to the rest of the samples, ranging from 10–22 nm. A wide range of sizes were produced in the case of mint. In Figure 4, lemongrass- and coriander-based nanoparticles ranged from 10–50 nm and 5–37 nm, respectively, which is shown clearly in Figure 4.

### 3.5. Thermogravimetric Analysis Curve

The capping agent and other organic impurities could function as contaminants in the final sample, decreasing the conductivity. Therefore, TGA was used to make sure they were taken out of the product. TGA was thus carried out both before and after the washing process. TGA was done to confirm the purity of Ag-NPs and to wash samples that initially included polymeric contents. Figure 5 shows the presence of pure Ag-NPs, as only one transition of Ag appeared. The slight increase in weight is due to oxidizing of Ag under an O_2_ environment.

### 3.6. XRD Analysis

To analyze the crystallographic structure of the Ag-NPs, XRD analysis was performed. Ag-NPs produced showed an FFC crystal structure. Figure 6 shows an XRD graph showing characteristic peaks at 38, 44, 64.5, and 77.5° (2θ value). These peaks were indexed as (111), (200), (220), and (311), respectively [36,37]. In our results, the peaks are at 38.1, 44.8, 64.5, and 77.5, which is very close. Since the samples were characterized on a thin glass film, this might be the factor that caused the peaks to show a minor shift.

The crystalline structure of biologically synthesized Ag-NPs using *Aloe vera*, coriander, mint, and lemongrass extract was analyzed via XRD measurements. A typical XRD pattern of the Ag was found by Bragg reflections corresponding to (111), (200), (220), and (311) sets of lattice planes that could be indexed on the bases of the FCC structure of silver. The weak intensities of peaks indicate that the silver nanoparticles were placed on a thin film for the XRD measurements.

### 3.7. Comparative Study of Mung Bean (Vigna radiata L.) Responses towards Ag-NPs and Their Salt

The application of Ag-NPs and AgSO_4_ on mung bean seedlings was investigated in this study. Ag-NPs and AgSO_4_ concentrations of 10 mg/L and 50 mg/L were compared to a control (0 mg/L). Subsequent effects on plant morphology were observed for three consecutive days in three replicates. Both concentrations caused noticeable effects under constant external conditions, which are shown in Figure 7 and Figure 8, respectively.

It was observed that Ag-NPs and AgSO_4_ treatments promoted root growth to a maximum root length at a concentration of 50 mg/L on day 1 and day 3. Lower concentrations boosted root growth compared to higher salt concentrations, and we concluded that Ag-NPs were more effective than salt treatment (A).

In B, Ag-NPs and AgSO_4_ treatments enhanced shoot growth to a maximum shoot length of 50 mg/L NPs on day 3 and higher salt concentrations showed deterioration. This shows that Ag-NPs were more effective than salt treatments. 

In Figure 7C, the calculated F value is 0.6760, whereas tabulated F (4,15) is 3.06. Compared to a 5% level for significance, we obtained a *p*-value of 0.6190. Since the critical value is larger than α = 0.05, the null hypothesis of equal population means was accepted, and we concluded that the results are insignificant at this level. In Figure 7D, the F value is 0.4725, whereas tabulated F (4,15) is 3.06. Compared to a 5% level for significance, we obtained a *p*-value of 0.7553. Since the critical value is larger than α = 0.05, the null hypothesis of equal population means was accepted, and we concluded that the results are insignificant at this level.

A significant increment in root weight was observed in treated plants. Higher concentrations showed more marked variations in biomass than lower concentrations. The maximum root weight was observed for the 50 mg/L NPs group on day 1 and day 3. On Day 2, the 50 mg/L AgSO_4_ group showed the maximum value (A). Increased shoot weight was observed at higher concentrations of NPs and salt. Higher biomass values were recorded at 10 mg/L of Ag-NPs compared to 10 mg/L of Ag SO_4_. On day 3, the maximum shoot weight was observed in the 50 mg/L NPs groups. In Day 2 and Day 3, the 10 mg/L AgSO_4_ group showed constancy in biomass (B). 

Figure 8C shows a calculated F value of 0.3823, whereas tabulated F (4,15) was 3.06. Compared to a 5% level for significance, we obtained a *p*-value of 0.8178. Since the critical value is larger than α = 0.05, the null hypothesis of equal population means was accepted, and we concluded that the results are insignificant at this level. Figure 8D shows a calculated F value of 0.9777, whereas tabulated F (4,15) was 3.06. Compared to a 5% level for significance, we obtained a *p*-value of 0.4488. Since the critical value is larger than α = 0.05, the null hypothesis of equal population means was accepted, and we concluded that results are insignificant at this level.

### 3.8. In Vitro Susceptibility, Minimum Inhibitory Concentration (MIC), and Minimum Bactericidal Concentration (MBC) Evaluation of Ag-NPs

Ag-NPs accumulate within the cell membrane of bacteria, which results in the permeability of the cell membrane, causing cell death [38]. The antibacterial activities of biosynthesized Ag-NPs solution against *E. coli* and *S. aureus* were investigated using a disc diffusion method. Figure 9 and Table 1 show the ZOI for *E. coli* (1.9, 2.1, 1.7, and 2 mm) and *Staphylococcus aureus* (1.8, 1.7,1.6, and 1.9 mm) against coriander, mint, *Aloe vera*, and lemongrass. 

The disc diffusion test is indicated as the preliminary study for screening the antibacterial activity of an antimicrobial agent, but further evaluation to determine the antibacterial activity of Ag-NPs in terms of MIC and MBC was needed [39]. MIC was defined as the lowest concentration of the antibacterial agent that inhibited the growth of bacteria by serial dilution. The bactericidal activity of Ag-NPs showed relatively effective results in the case of *E. coli*, which is a gram-negative bacterial species, and *Staphylococcus aureus*, which is a gram-negative bacterium. Previous studies also show that gram-negative bacteria (e.g., *E. coli*) and gram-positive bacteria (e.g., *S. aureus*) show more sensitivity towards silver nanoparticles [40]. 

The MIC values of *E. coli* against coriander, mint, *Aloe vera*, and lemongrass ranged from 7.3 to 8 µg/mL while *Staphylococcus aureus* showed an MIC range of 7.2 to 7.9 µg/mL. Similarly, the MBC values of *E. coli* and *Staphylococcus aureus* ranged from 7.2 to 8.3 and 7.1 to 7.8 µg/mL, respectively. The MIC and MBC value of *E. coli* showed that *E. coli* was less susceptible to Ag-NPs. 

The antibacterial activity of Ag-NPs has been reported by many researchers. However, the MIC values from previous studies show large variation in their ranges. Therefore, the comparison of the results is difficult. As there is no standard method for the determination of the antibacterial activity of Ag-NPs, different methods have been applied by researchers [41].

## 4. Discussion

Plant extracts have now been increasingly used to combat multidrug resistance [31,42,43]. However, there is now a new dimension to tackling microbial invasion in the form of green nanotechnology. These green-synthesized silver nanoparticles, which were synthesized with plant extracts, can be used in drug delivery, proteomic studies, cancer therapy, and medicine. A facile approach to synthesizing silver nanoparticles from plant extracts is cost-effective as well as non-hazardous. The results show that the mean particle size of the synthesized nanoparticles was reduced with the addition of their good absorbance properties as well as their crystallinity. In the future, these nanoparticles can be used in proteomic studies and drug delivery applications.

Environmentally acceptable Ag-NP synthesis methods using plant extracts are far better candidates to produce Ag-NPs than physical, chemical, or microbiological techniques. Plants are more helpful than other biological agents, such as bacteria, as there is no requirement for plant culture upkeep. Plants provide a wealth of medicinally significant metabolites [44] that can also function as capping and reducing agents in the production of Ag-NPs [45].

Because of their small size and large specific surface area, biosynthesized Ag-NPs have a high adsorption capacity and are likely to form coordination bonds with bio-molecule groups such as phosphate, amino groups in proteins, carbonyl group, negative oxygen ions, thiol, and disulfide bonds in DNA (a cell membrane consists of two layers of phospholipid and inlaid DNA). 

Due to the growing consumption of Ag-NPs, there is an increased risk of their release into the atmosphere. They have accrued in the earth and water reservoirs in huge amounts and are unavoidably absorbed by crops, easily entering the food chain [46]. Therefore, the effects of synthesized nanoparticles on plants should be assessed before their utilization. 

In this study, the toxicity of the synthesized Ag-NPs was analyzed on mung bean seedlings. Significant alterations in plant morphology were found after exposure to Ag-NPs. Plant length and biomass were used for assessing the phytotoxicity of Ag-NPs. The NP application caused a significant increase in all parameters. The nanomaterials were applied in two different concentrations, whereas the same concentrations of the parent salt material were also used. This increase was found to be dose dependent. This increase was also found to be more pronounced in the case of nanoparticles as compared to the same concentration of the parent bulk material. Several studies have found a positive relationship between phytotoxicity and the concentration of Ag-NPs during exposure. The concentration of Ag-NPs only above the threshold level can cause negative effects. The morphology or structure of rice root, when exposed to 30 g/mL, remained unchanged, while, with an increased concentration of 60 g/mL, Ag-NPs stabbed the cell wall and disrupted cell morphology and structural features [47]. Root length was found to be enhanced at a 30 g/mL concentration, while 60 g/mL restricted root growth. Seedlings of Lupinus termis grown with high doses of Ag-NPs (300–500 ppm) had less shoot and root elongation as well as less biomass accumulation. Ag-NPs at 100 ppm also encouraged shoot and root growth [48]. In Pisum sativum, a low-concentrated Ag-NP solution was found to have beneficial impacts on seedling development [49].

It has been reported that in the aquatic plant *Lemna gibba*, Ag-NP treatment increased intracellular ROS generation, which was completely associated with the rising absorption of Ag-NPs [50]. Similarly, elevated Ag-NP concentrations in jasmine rice were observed to reduce seed germination following seedling development [51]. Ag-NPs induce oxidative stress and exhibit phytotoxicity only when introduced in higher concentrations in *Allium cepa* roots [52]. Exposure to higher concentrations of Ag-NPs caused a decrease in the biomass in *Arabidopsis* [53]. Ag-NPs also decrease the length of shoots and roots of wheat in a dose-dependent manner [54]. Other research involving numerous plant species, including Arabidopsis, found similar findings regarding Ag-NP toxicity on seed germination, biomass accumulation, and root and shoot growth [55], *Brassica nigra* [56], *Lemna* [57], *Phaseolus radiatus, Sorghum bicolor* [27], *Lolium multiflorum* [58], rice [59], wheat [60], and *Lupinus termis* L [61]. Due to their exceptional antibacterial qualities, silver nanoparticles (Ag-NPs) are widely used in a variety of industries. However, numerous investigations have shown that Ag-NPs have the potential to be cytotoxic due to their impact on the excessive generation of reactive oxygen species (ROS) in various cells and pathological conditions like cancer [62]. Ag-NPs have strong antimicrobial properties against many fungi, viruses, and bacteria due to their activity as photocatalysts and their ability to produce reactive oxygen species [1].

## 5. Conclusions

Ag-NPs are used in many different industries in modern civilization, due to which their dissemination and absorption into the ecosystem is unavoidable. Therefore, understanding the transfer of Ag-NPs throughout the ecosystem and their impacts on plants is of crucial importance. The present investigation showed that Ag-NPs possess strong antibacterial activity and that their application positively affected plant growth in mung bean seedlings. Most of the studies conducted so far have reported positive effects of Ag-NPs on plant growth and progress. However, some studies have also shown harmful effects of Ag-NPs on plants in different aspects. These contradictory results indicate the complexity of the responses in plants to Ag-NPs; responses which are dependent on the properties of Ag-NPs, the plant used, and the mode of exposure.

## Data Availability

The authors confirm that the data supporting the findings of this study are available within the article.

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
