# Peer review of "Phyto-Synthesis, Characterization, and In Vitro Antibacterial Activity of Silver Nanoparticles Using Various Plant Extracts"

_bioengineering, 2022, doi:10.3390/bioengineering9120779_

Round 1

Reviewer 1 Report (Previous Reviewer 1)

This manuscript has been greatly improved after revision in many parts.  It can be published as its current form.

Author Response

We are very thankful for your valuable comments, suggestions, and recommendation. We carefully checked the English language and grammar and corrected the manuscript accordingly. 

Reviewer 2 Report (Previous Reviewer 2)

In my opinion, the publication is not suitable for publication. The results of bactericidal activity are inconsistent and mutually exclusive.

Author Response

All the authors are highly grateful to you for spending time and giving valuable comments that we believe surely improved our manuscript. As suggested, we have provided adequate background and relevant references in the introduction section. The MIC and MBC tests were also performed according to your suggestions and recommendations. The research design of these tests is elaborated in the Materials and method section and results are also clearly described in the Results section. The conclusion of these results is described properly in the Discussion section which supports our results and methods accordingly.

Reviewer 3 Report (Previous Reviewer 3)

The authors have nicely revised the original manuscript.

In the present form, the manuscript is ready for publication as it is.

Author Response

The authors are thankful to you for your valuable comments and recommendations. The manuscript has been revised carefully for further consideration. 

This manuscript is a resubmission of an earlier submission. The following is a list of the peer review reports and author responses from that submission.

Round 1

Reviewer 1 Report

In this study, the authors synthesized stable silver nanoparticles (Ag- NPs) from Aloe Vera, Mangifera indica (mango), Coriandrum sativum (coriander), and Cymbopogon citratus (lemongrass) leaf extract. This is the first work reported that includes the synthesis of silver nanoparticles by controlling the pH so that the particle size can be reduced, making this method not only cost-effective but also increasing the properties of nanoparticles. Here are a few minor concerns which are suggested to be considered: 

1. Can the author explain why all the Ag synthesized in Fig2 are agglomerated?

2. The peak in Figure 6 does not match, please improve it.

3. Error bar and significance differences should be added in Figure 7 and 8.

Author Response

Why the Ag synthesized in Fig 2 are agglomerated?

Ans: Thank you for your recognition of our work and valuable comments. The Ag synthesized in this research work was in the form of suspended nanoparticles in DI ethanol. The SEM sample preparation was done without sonication to minimize the risk of impurities. The agglomeration is due to the adhesion of particles to each other by weak forces once the samples are prepared on carbon tape. The stages of agglomeration are particle movement, particle collision, particle attachment, and either disintegration or cementation of the attached particles.

The Peak in Fig 6 does not match?
Ans:
XRD of silver shows 4 peaks. These are (111), (200), (220) and (311) at 2θ value of 38.29, 44.55, 64.81 and 78.05 according to 04-0783 JCPD card number. In our results, the peaks are at 38.1, 44.8, 64.5, and 77.5 which is very close. Since the samples were characterized on a thin glass film, this might be the factor that the peaks show a minor shift.

Error bar and significance difference should be added in figure 7 and 8.

Ans: Figure 7 and 8 have been improved by added the error bars and significance. The description has been improved as well.

Reviewer 2 Report

Basically, the approach taken in the study is interesting, but data is not appropriate for publication in Bioengineering. My opinion is based on the following doubts and questions. 1. Section Introduction: The introduction section is too long and off topic. Why do the authors write about TiO2 NPs if the article concerns silver nanoparticles? Do not write Gramme positive or Gramme negative only Gram-positive and Gram-negative. Instead of writing about the plants used for the synthesis of AgNPs, we should mainly mention the secondary metabolites contained in these plants, which influence the synthesis of AgNPs. 2. Section Materials and Methods: The section on the synthesis of AgNPs was completely omitted. Scheme 33 is not enough for the synthesis to be repeated. The 3-day AgNPs toxicity study on mung bean seedlings is insufficient. The experiment time is too short to draw correct conclusions. The authors wrote that measurements were taken after days 7, 8 and 9 of the experiment, but the results are only for days 1, 2 and 3 of the experiment. The accumulation of AgNPs in plants has not been investigated. Since mung beans are consumed by humans, the first thing to check is whether AgNPs are not accumulating in the plant. 3. Section Results and Discussion: I have the most doubts about the results concerning the antibacterial activity of the obtained silver nanoparticles. In Fig. 9, I do not see the zones of growth inhibition described in Table 1. In my opinion, the silver nanoparticles are absorbed into the paper discs and are inactive. To check the antimicrobial activity, a test should be performed according to CLSI guidelines for minimal bactericidal concentration (MBC) determination (CLSI 1996). What does 10%, 30% and 50% mean in Figure 9? Why is there no antibiotic and AgNO3 control in Figure 9? Figure 9 is illegible. I do not agree with the authors' statement in the conclusions section that the AgNPs obtained by them show strong antibacterial activity.

First of all, the main objection is the fact that the synthesis of silver nanoparticles using plants such as Aloe vera, Coriandrum sativum, Mangifera indica and Cymbopogon citratus has already been shown many times in the scientific literature. I do not see any signs of novelty here. Worse, the authors did not properly discuss their results with the literature data. In the Discussion section, the authors wrote the following quote: “This is the first work reported that includes the synthesis of silver nanoparticles by controlling the pH so that the particle size can be reduced, making this method not only cost-effective but also increasing the properties of nanoparticles.

Unfortunately, in the manuscript the word pH actually only appears in the description of Scheme 33. Unfortunately, in the manuscript the word pH actually only appears in the description of Scheme 33. Have you studied the formation of AgNPs at a different pH? If not, why was this pH chosen? In my opinion, the conclusions section completely misses the content of the manuscript. In the discussion section, the authors devote themselves to the activity of AgNPs without examining their activity beforehand [e.g. “Because of their small size and large specific surface area, biosynthesized Ag-NPs have a high adsorption capacity and are likely to form coordination bonds with bio-molecule groups such as phosphate, amino groups in proteins, carbonyl group, negative oxygen ions, thiol, and disulfide bond in DNA (a cell membrane consists of two layers of phospholipid and inlaid DNA).”].

In summary, the work submitted for review is another work that does not bring anything new to the world of science and should not be published in Bioengeneering.

Reviewer 3 Report

Manuscript ID: bioengineering-1847766
Type of manuscript: Article
Title: Rapid biosynthesis, characterization and bioactivity of Silver
nanoparticles using various Plant Extracts
Authors: Bilal Ahmad *, Usama Qamar Satti, Sami ur Rehman, Huma Arshad,
Ghazala Mustafa, Uzma Shaukat, Chunyi Tong *

Ag-NPs were synthesised by leaf extract from Aloe Vera, Mangifera indica (mango), Coriandrum sativum (coriander), and Cymbopogon citratus (lemongrass).  Several techniques, such as UV–vis spectrophotometer, XRD,TGA), SEM), and energy dispersive X-ray spectroscopy have been used to characterise the Ag-NPs, which exhibited significant antibacterial activity, against E. coli.

I have major amendments;

- Revise the paper according to the style of MDPI, Bioengineering Journal;
- Improve the UV-vis discussion, it is poor and bad presented (samples are not named!).
- Improve ALL SEM image quality and related discussion.
- As well as the XRD (This is really bad reported!)
-Up to now the manuscript seems a list of technical reports, bad described. Please harmonize the text, being precise and referring better to literature. 

It is strange, there is a Results and discussion paragraph, and after a Discussion.

Please improve the manuscript, which it is interesting, but in the present form it is unpublishing in Bioengineering Journal.

Author Response

Manuscript: Rapid biosynthesis, characterization and bioactivity of Silver nanoparticles using various Plant Extracts

Manuscript ID: bioengineering-1847766

Authors: Bilal Ahmad1*, Li Chang1, Usama Qamar Satti2, Sami-ur-Rehman3, Huma Arshad4, Ghazala Mustafa4, Uzma Shaukat5 and Chunyi Tong1*

Revise the paper according to the style of MDPI, Bioengineering journal.

Thank you for your recognition of our work and valuable comments. The paper has been revised according to the MDPI style.

Improve UV-Vis Discussion, it is poor and bad presented. (Samples are not named).

The kinetics of reduction of aqueous silver ions during reaction with the leaf extracts broth may be easily followed by UV-vis-NIR spectroscopy. It is well known that silver nanoparticles exhibit reddish brown colors, these colors arising due to excitation of surface plasmon resonance (SPR) in the silver nanoparticles. Figure 1 shows the UV-vis-NIR spectra recorded from the aqueous silver leaf extracts of mint, aloe vera, coriander and lemongrass broth reaction medium as a function of time of reaction. It is observed that as the reaction proceeds and the pH of the solution is increased to 9, the silver SPR band at ca. 390-470 nm for the leaf extracts broth steadily decreases in intensity. This band is indicative of the presence of nanoparticles in solution. In addition to the peaks in the abovementioned range, a progressive decrease in the absorption at smaller wavelengths into the near-infrared (NIR) region of the electromagnetic spectrum is observed.

Improve SEM image quality and related Discussion

The SEM analysis clearly showed the presence of the synthesized Ag-NPs. NPs ranging from 15 to 50 nm were observed in the surface morphology of AuNPs. It was noted that smaller sized particles were almost spherical in shape, and some of them were aggregated (Figure 2). The SEM micrographs showed aggregates of synthesized Ag-NPs, and the particles were in the range of 15–25 nm and are in direct contact even within the aggregates, indicating the stability of NPs by controlling the pH.

As well as the XRD (this is really bad reported).

The crystalline structure of biologically synthesized Ag NPs using Aloe Vera, coriander, mint, and lemongrass extract were analyzed by XRD measurements. A typical XRD pattern of the Ag was found by Bragg reflections corresponding to (111), (200), (220) and (311) sets of lattice planes are observed that may be indexed on the bases of the FCC structure of silver. The characteristic peaks corresponding to (111), (200), (220) and (311) are located at 2θ = 38.1, 44.8, 64.5, and 77.5 which is very close at 2θ value of 38.29, 44.55, 64.81 and 78.05 according to 04-0783 JCPD card number of silver nanoparticles respectively. The weak intensities of peaks indicates that silver nanoparticles are placed on a thin film for the XRD measurements.

Up to now the manuscript seems a list of technical report, bad described. Please harmonize the text, being precise and referring better to literature.

The overall manuscript has been thoroughly revised for flow and consistency.

It is strange, there is results and discussion paragraph, and after a discussion.

It was typographical error and corrected in the main manuscript.

Round 2

Reviewer 2 Report

In the AgNPs synthesis schematic diagram (page 4) the authors noted that they kept AgNPs at 4 degrees Celsius and below the diagram it says that they kept AgNPs at 40 degrees Celsius. How were AgNPs stored? To my knowledge, AgNPs should not be kept in a refrigerator because large aggregates precipitate.

Unfortunately, the antimicrobial tests are still poorly done. I asked to
have the MIC and MBC determined. The authors made circles on the
photos where they marked the zone of inhibition of growth. The values
​​
given by the authors in millimeters of growth inhibition are not
consistent with what can be seen in the poor quality photos.
Additionally, in Figure 9 it is not known what kind of AgNPs are tested.
The authors obtained 3 types of AgNPs from three different plant
species. So which AgNPs are in Figure 9?
To me, 10%, 30% and 50% of AgNPs are completely unclear
(Fig. 9, Table 1). How is it possible that with increasing amounts of
AgNPs the zone of inhibition of bacterial growth decreases?
In fact, the difference between 8, 10 and 12 millimeters is not a
difference and it is statistically insignificant. In my opinion, the activity
is minimal. Without proper antimicrobial testing, no real conclusions
can be drawn.

So I completely disagree with the authors' conclusions that:
“Based on these results it can be concluded that the synthesized
Ag-NPs exhibited significant antibacterial activity and improved
plant growth”.